# FAST AND SLOW LEARNING OF RECURRENT INDEPENDENT MECHANISMS

**Kanika Madan[1], Rosemary Nan Ke [2], Anirudh Goyal [1], Bernhard Schölkopf [3], Yoshua Bengio [1]**

## ABSTRACT

Decomposing knowledge into interchangeable pieces promises a generalization advantage when there are changes in distribution. A learning agent interacting with its environment is likely to be faced with situations requiring novel combinations of existing pieces of knowledge. We hypothesize that such a decomposition of knowledge is particularly relevant for being able to generalize in a systematic way to out-of-distribution changes. To study these ideas, we propose a particular training framework in which we assume that the pieces of knowledge an agent needs and its reward function are stationary and can be re-used across tasks. An attention mechanism dynamically selects which modules can be adapted to the current task, and the parameters of the *selected* modules are allowed to change quickly as the learner is confronted with variations in what it experiences, while the parameters of the attention mechanisms act as stable, slowly changing, meta-parameters. We focus on pieces of knowledge captured by an ensemble of modules sparsely communicating with each other via a bottleneck of attention. We find that meta-learning the modular aspects of the proposed system greatly helps in achieving faster adaptation in a reinforcement learning setup involving navigation in a partially observed grid world with image-level input. We also find that reversing the role of parameters and meta-parameters does not work nearly as well, suggesting a particular role for fast adaptation of the dynamically selected modules.

## 1 INTRODUCTION

The classical statistical framework of machine learning is focused on the assumption of independent and identically distributed (i.i.d) data, implying that the test data comes from the same distribution as the training data. However, a learning agent interacting with the world is often faced with non-stationarities and changes in distribution which could be caused by the actions of the agent itself, or by other agents in the environment. A long-standing goal of machine learning has been to build models that can better handle such changes in distribution and hence achieve better generalization (Lake & Baroni, 2018; Bahdanau et al., 2018). At the same time, deep learning systems built in the form of a single big network, consisting of a layered but otherwise monolithic architecture, tend to co-adapt across different components of the network. Due to a monolithic structure, when the task or the distribution changes, a majority of the components of the network are likely to adapt in response to these changes, potentially leading to catastrophic interferences between different tasks or pieces of knowledge (Andreas et al., 2016; Fernando et al., 2017; Shazeer et al., 2017; Jo et al., 2018; Rosenbaum et al., 2019; Alet et al., 2018; Kirsch et al., 2018; Goyal et al., 2019; 2020; Goyal & Bengio, 2020).

An interesting challenge of current machine learning research is thus out-of-distribution adaptation and generalization. Humans seem to be able to learn a new task quickly by re-using relevant prior knowledge, raising two fundamental questions which we explore here: (1) how to separate knowledge into easily recomposable pieces (which we call modules), and (2) how to do this so as to achieve fast adaptation to new tasks or changes in distribution when a module may need to be modified or when different modules may need to be combined in new ways. For the former objective, instead of representing knowledge with a homogeneous architecture as in standard neural networks, we adopt recently proposed approaches (Goyal et al., 2019; Mittal et al., 2020; Goyal et al., 2020; Rahaman

---

[01] Mila, University of Montreal, [2] Mila, Polytechnique Montréal, [3] Max Planck Institute for Intelligent Systems. Corresponding author: `madankanika.s@gmail.com`

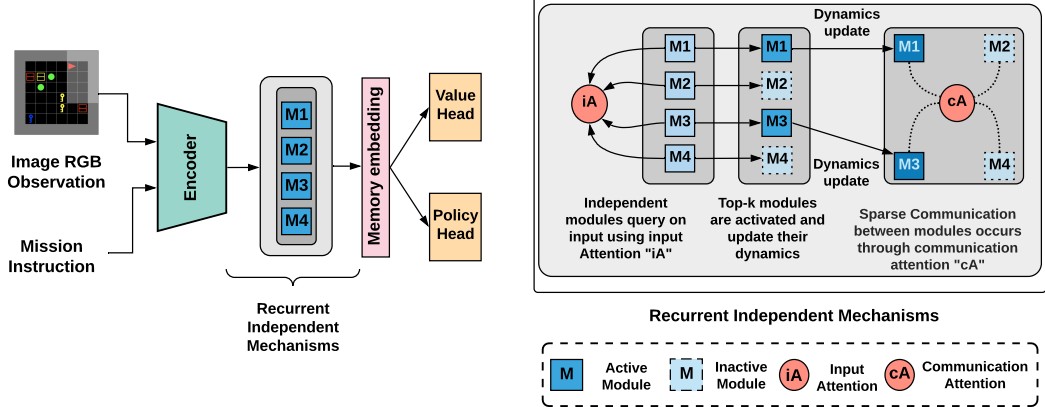

Figure 1: **Proposed Model architecture:** Input images processed through an encoder, and the embedded mission instruction are passed through a set of recurrent independent modules or RIMs (Goyal et al., 2019) which compete using attention mechanisms to capture the dynamics of the system. The network's state is divided into $N$ modules, such that at every time step, conditioned on a given input $x_t$, only a subset $k$ out of total $N$ modules are dynamically activated and updated fast in the inner learning loop, while the parameters of the two attention mechanisms, Input Attention and Communication Attention, are learnt slowly in the outer loop as meta parameters.

et al., 2020) to learn a modular architecture consisting of a set of *independent modules* which compete with each other to attend to an input and sparsely communicate using a key-value attention mechanism (Bahdanau et al., 2014; Vaswani et al., 2017; Santoro et al., 2018). For the second objective of fast transfer, we adopt a meta-learning approach on the two sets of parameters (those of the modules and those of the attention mechanisms) with the goal of achieving fast adaptation to changes in distribution or a new task in a reinforcement-learning agent (Duan et al., 2016; Mishra et al., 2017; Wang et al., 2016; Nichol et al., 2018; Xu et al., 2018; Houthooft et al., 2018; Kirsch et al., 2019).

In this paper, we study the generalization ability of the proposed modular architecture on tasks not seen during training. We conduct experiments in the domain of grounded language learning, in which poor data efficiency is one of the major limitations for agents to learn efficiently and generalize well (Hermann et al., 2017; Chaplot et al., 2017; Wu et al., 2018; Yu et al., 2018; Chevalier-Boisvert et al., 2018). We show, empirically, how the proposed learning agent can generalize better not only on the seen data, but also is more sample efficient, faster to train and adapt, and has better transfer capabilities in the face of changes in distributions. We thus present evidence that combining meta-learning with modular architectures can help in building agents that learn and leverage the compositional properties of the environment to generalize better on novel domains and achieve better transferability and a more systematic generalization.

## 2 META LEARNING OF RECURRENT INDEPENDENT MECHANISMS

We intend to assay whether a modular architecture, combined with learning different parts of the model on different timescales, can help in decomposing knowledge into re-usable pieces such that the resulting model is not only more sample efficient, but also generalizes well across changes in task distributions. We first give a high-level overview, and then describe the components of the model and the two learning phases in more detail: Section 2.1 contains an overview of the modular architecture consisting of an ensemble of recurrent modules, and Section 2.2 explains the meta-learning approach to learn the parameters of the modular network at different time scales.

***Modular Network.*** The proposed method is based on RIMs architecture (Goyal et al., 2019) which consists of a set of competing modules, such that each module acts independently and interacts with other modules sparingly through attention.

***Attention Networks to modulate information.*** The soft-attention mechanisms control the flow of information in the model such that different modules attend to different parts of the input via *input attention*, and a module queries relevant contextual information from other modules via *communication attention*. The two attention mechanisms can use multiple attention heads and have their own set of parameters, and the different modules have their own independent parameters.

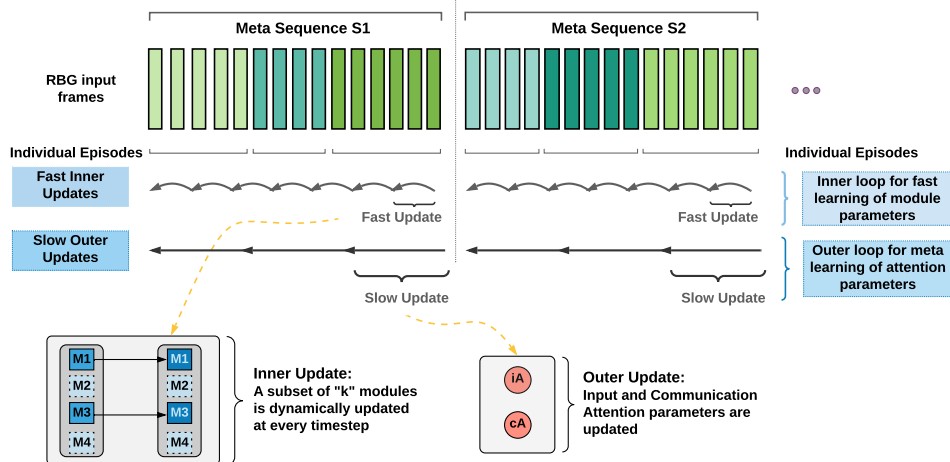

Figure 2: **Meta-Attention Setup:** The two learning loops (fast and slow) of the meta-learning setup learn different parameters of the model at different timescales. Several episodes are concatenated to form long sequences, $S1, S2, \ldots$ The learning happens over these meta sequences such that at every time-step, conditioned on the current input $x_t$, a dynamic subset of $k$ (out of total $N$) modules is learnt over shorter time-spans to capture the quickly changing dynamics, while the two sets of attention mechanisms, which define the connectivity structure and sparse communication between the modules, are learnt over much longer time-spans to capture more stable connectivity structures.

***Learning at different time scales.*** In the proposed model, the entire network is trained end-to-end using a meta-learning setup such that different components of the network are trained at different time scales and some parameters adapt quickly, while others are updated much less frequently.

***Fast learning of parameters of relevant modules.*** In order to capture the quickly changing aspects of the underlying task distribution, the parameters of the activated modules (most relevant to the current input) are updated in a fast manner by learning over multiple short spans of the input sequence.

***Slow learning of stationary meta-parameters of attention mechanisms.*** In order to capture the more stable aspects of the task distribution, the parameters of the two sets of attention mechanisms are updated less frequently over relatively longer spans of the input sequence. These attention mechanisms are responsible for the activation of the relevant modules and for the modulation of an information exchange between different modules. The parameters of the individual modules are not updated in this phase.

## 2.1 RECURRENT INDEPENDENT MECHANISMS

In this section, we describe the parameterization of the modular network, as well as how attention is used to select relevant modules at a particular time-step.

***Recurrent Independent Modules.*** In Meta-Attention Networks, we follow a similar modular architecture as in (Goyal et al., 2019), such that the network consists of an ensemble of independently parameterized modules in which each module operates with its own individual dynamics and these modules sparingly interact with each other through attention. At a given time-step $t$, for each of the $j = 1, \ldots, N$ modules, let $h_{t,j}$ and $D_j$ represent the hidden state vector and the recurrent dynamics, respectively, of the module $j$. Given an input $x_t$ at time $t$, the hidden states of the independent modules follow the following three-step process: First, based on their relevance to the current input $x_t$, a subset of $k$ (out of $N$) modules is dynamically activated to form an active set $A_t$ consisting of the most relevant modules. Second, these activated modules (in active set $A_t$) independently process the input using their *default* transition dynamics, $D_j$ for $j = 1, \ldots, k$, while the hidden state remains the same for the other $N - k$ non-activated modules. In the third step, the modules sparsely communicate with the other modules through an attention mechanism.

Two sets of attention mechanisms, (1) Input Attention and (2) Communication Attention, labeled as "iA" and "cA" respectively in Fig. 1, are used to selectively activate modules and to modulate a sparse communication between them. We explain these two attention mechanisms below.

***Soft attention mechanisms.*** Attention mechanisms used for selective activation of different modules (Input Attention) and for communication between the modules (Communication Attention) are based on multi-headed soft attention. The attention is computed as $\text{Attention}(Q, K, V) =$

softmax $\left(\frac{QK^T}{\sqrt{d_k}}\right) V$, where $Q$, $K$ and $V$ represent the query, key and value, and $d_k$ is the size of the features $K$. The query is obtained as a function of where the information is written to, while keys and values are obtained as a function of where the information is read from.

***Input Attention for selective activation of competing modules.*** At every time step $t$, each module generates a query, and keys/values are obtained as a function of the input. Then a soft key-value based attention mechanism (Bahdanau et al., 2014; Vaswani et al., 2017; Santoro et al., 2018) is used to select the top-$k$ modules that pay most attention to input $x_t$ in a differentiable way using a sparse softmax (Ke et al., 2018; Goyal et al., 2019). Using this style of competition between different modules has shown to improve specialization among different modules (Goyal et al., 2019; 2020).

***Communication Attention for sparse communication between modules.*** In order to share information between different modules, modules communicate with each other through a second attention mechanism, called Communication Attention. During this process, only the activated modules in $A_t$ can read relevant contextual information from all the other modules, while the non-activated modules remain unchanged. This process is also based on soft key-value based attention (Bahdanau et al., 2014; Vaswani et al., 2017). Here, each module in the active set $A_t$ makes a query, while keys and values are a function of other modules (irrespective of if they are active or not).

***Hyperparameters.*** In the proposed method, the values of $k$ and $N$ are hyper-parameters that stay constant throughout the learning of a particular task as well as across different environments.

## 2.2 META-LEARNING ATTENTION TO MODULATE INFORMATION BETWEEN MODULES

In order to appropriately capture the quickly vs. slowly changing aspects of the underlying distribution, we use a meta-learning setup such that not all the parameters are trained at the same pace: some parameters are adapted more quickly than others and the updates happen in their respective loops only. Throughout the meta-learning process, the input observation sequence from several concatenated episodes is divided into shorter and (relatively) longer horizons, as visualized in Fig. 2, and the learning of the full set of network parameters happens over the following two update loops:

***Inner / Faster loop.*** Conditioned on the input $x_t$ at time $t$, the parameters of the $k$ most relevant modules in active set $A_t$ are updated over several short spans of the input sequence, while keeping the parameters of the attention mechanisms constant. Given the dynamic selection of the $k$ most relevant modules, different sets of modules may get updated over different iterations of this update loop.

***Outer / Slower loop.*** In the outer loop, longer spans of input observations are considered to slowly update the parameters of the attention mechanisms, while the parameters of the independent modules are kept unaltered. This captures the more stable aspects and connectivity patterns across the network.

***Loss Function.*** For our experiments, we used the Proximal Policy Optimization algorithm (Schulman et al., 2017) using parallel data collection. During the training of the parameters of the network, we use separate output heads for the policy and value function. These two heads are updated separately on different learning time scales, such that the value function head is updated slowly in the outer meta loop along with the attention parameters to capture regularities in the environment, while the policy head is updated in a fast way in the inner loop along with the parameters of the selected recurrent modules. In each loop, only the respective parameters specified for that loop are updated while keeping others unaltered, and the outer loop gradient is not obtained by backpropagating through the inner loop updates. For this work, the number of steps through which the gradients are back-propagated in time in the outer loop is four times longer than in the inner loop, making the outer loop sweep across four times longer time-steps as compared to the faster inner loop.

## 3 RELATED WORK

**Meta-Learning:** Meta-learning (Bengio et al., 1990; Schmidhuber, 1987) methods gives the flexibility to adapt to new environments rapidly with a few training examples, and has demonstrated success in both supervised learning such as few shot image classification (Ravi & Larochelle, 2016) and reinforcement learning (Wang et al., 2016; Santoro et al., 2016; Duan et al., 2016; Botvinick et al., 2019; Clune, 2019) settings. The most relevant modular meta-learning work is that of Alet et al. (2018), which proposes to learn modular network architecture based on MAML, however their approach relies on pre-trained composable transformations. The goal of the current work is to

learn the transformations (i.e decomposition of knowledge into separate modules), as well as how to dynamically route information among such modules.

**Meta-Learning to Disentangle Causal Mechanisms:** Recently (Bengio et al., 2019; Ke et al., 2019) used meta-learning to learn causal mechanisms or causal dependencies between a set of high-level variables, that inspired the approach presented here. The 'modules' in their work are the conditional distributions for each variable in a directed causal graphical model (Schölkopf et al., 2016). The inner-loop of meta-learning also allows the modules to be adapted within an episode (corresponding to an intervention distribution), while the outer-loop of meta-learning discovers how the modules are connected (statically) to each other to form the graph structure of the graphical model.

**Modular Networks.** The generalization capabilities of neural models may benefit from structure in the design of models, to make them resemble the kind of rules they are supposed to learn (Andreas et al., 2016; Johnson et al., 2017; Jacobs et al., 1991; Bottou & Gallinari, 1991; Ronco et al., 1997; Reed & De Freitas, 2015; Andreas et al., 2016; Rosenbaum et al., 2017; Fernando et al., 2017; Shazeer et al., 2017; Kirsch et al., 2018; Rosenbaum et al., 2019). These networks has an architecture which is composed dynamically from several neural modules, where each module is meant to perform a distinct function. In such architecture different modules are applied one at a time, in contrast to the proposed architecture, where different modules can update their state and be used for prediction in parallel. The focus of this work is to automatically decompose knowledge into a set of independent modules as well as learn input-dependent connectivity between these modules. We show that using a meta-learning setup helps in learning better decomposition of knowledge such that the resulting model can generalize better out-of-distribution scenarios.

## 4 EXPERIMENTS

We evaluate the proposed Meta-RIMs networks to answer the following questions: (a) Does the proposed method improve sample efficiency? We answer this positively in section 4.1. (b) Does the proposed method lead to policies that generalize better to systematic changes to the training distribution? We find positive evidence for this in section 4.2 (c) Does the proposed method lead to a faster adaptation to new distributions and a better curriculum learning regime to train agents in an incremental fashion by reusing the knowledge from previously learnt similar tasks? We evaluate this setting and find positive evidence in section 4.3. We also conduct ablation studies to understand the importance of different components of the proposed setup (attention-controlled modularity and meta-learning) and these are summarized in section 4.4.

**Environments:** The experiments are based on a large variety of environments from the MiniGrid and BabyAI suite (Chevalier-Boisvert et al., 2018) which provide an egocentric and partially observed view of the environment. Partial observability, sparse rewards, and a procedurally generated series of environments with a systematically incremental difficulty make faster learning particularly challenging for reinforcement learning agents, but make it useful to address the questions we raised above. Throughout the training, the agent is presented with sparse rewards, with a positive reward given only when the goal is reached within a certain number of timesteps. We use the metrics of mean reward $R$ and average success-rate $S$ (i.e. the mean of the rewards obtained and the mean of the percentage of times the agent succeeds in achieving the goal, across multiple runs) in the results, showing the learning curves (as a function of number of input frames). Please refer to Appendix A.1 for additional details on the environments and hyperparameters used.

**Baselines:** We compare the performance of Meta-RIMs Networks with the following baselines in which all the parameters of the network are learnt at the same time without any meta-learning setup: (a) A *Vanilla LSTM* model (b) A modular network (i.e RIMs (Goyal et al., 2019)). These are referred to as "metaRims", "vanilla" and "modular", respectively, in the following sections.

### 4.1 IMPROVED SAMPLE EFFICIENCY

Training agents in a sample efficient manner is one of the major challenges in reinforcement learning. Training different parameters of the model at different timescales in the proposed setup leads to a more sample efficient training regime, as evident across a wide range of BabyAI and MiniGrid tasks (GoToLocal, PickupDist, GoToRedBall, DoorKey, PutNear, Fetch, FourRoomsS13, GoToObj, MemoryS13Random), see Fig. 3 for Mean Reward $R$, and Fig. 4 for Mean Success Rate $S$.

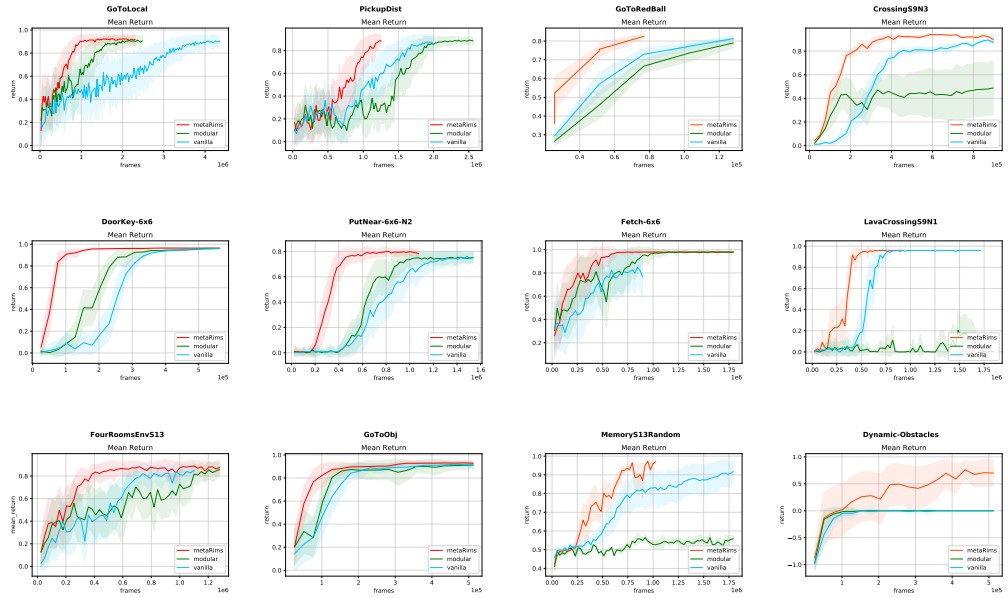

Figure 3: **Sample Efficiency on MiniGrid and BabyAI environments:** proposed method (labeled "metaRims") with modular architecture and meta-learning (red), outperforms the "modular" RIMs (green) and "vanilla" LSTM (blue) baselines. Improvements (in mean-return vs number of frames seen) are more profound in more difficult environments.

## 4.2 BETTER POLICY GENERALIZATION

We evaluate the capability of the proposed model to transfer knowledge from an easy environment to similar but incrementally more difficult environments, such that the environments share most aspects of the underlying structure. More specifically, the agent has to find a key to unlock a door and reach the goal in the other end of the room. As the environments get progressively more difficult, the rooms become larger leading to even longer trajectories and sparser rewards, see Fig. 10 in Appendix. In the scenario of zero-shot generalization to harder environments, the proposed "meta" method outperforms the other two baselines by a substantial margin in the more difficult environment, see Table 1. We found the three methods (meta, modular, and vanilla) to not degrade much in the backward direction.

| Target Environment | R(MetaRims) | R(Vanilla) | R(Modular) | S(MetaRims) | S(Vanilla) | S(Modular) |
|---|---|---|---|---|---|---|
| Difficult | 0.97 | 0.95 | 0.85 | 1.0 | 1.0 | 0.89 |
| More Difficult | 0.45 | 0.18 | 0.01 | 0.47 | 0.19 | 0.02 |

Table 1: **Zero shot Policy Transfer:** The model is trained on the easiest environment, and transferred in a zero-shot manner to a more difficult and larger environment, outperforming the baselines in terms of both rewards (R) and success rates (S) as the difficulty of environment increases.

## 4.3 EFFICIENT PRE-TRAINING AND KNOWLEDGE TRANSFER FOR CURRICULUM LEARNING

In order to evaluate how efficiently the proposed setup can leverage the previously learnt knowledge, we trained an agent on a source environment and used it as a pre-trained model to adapt to a target environment. To allow a reuse of knowledge across tasks, the source and the target environments share some underlying structure in the form of a set of core competencies, namely "Room", "Distr-Box", "Distr". As evident in Fig. 5(a), we find that the proposed model adapts much better as compared to the regular LSTM setup, indicating a more efficient reuse of the past experience.

## 4.4 ABLATION AND ANALYSIS

***Benefits of Meta-Learning Setup.*** In order to understand the impact of the proposed meta-learning setup and training different parameters at different timescales, we trained a vanilla LSTM baseline using a similar meta-learning setup, in which the parameters of the LSTM and the policy head are updated more often in the inner loop, while the parameters of the value function head are updated in

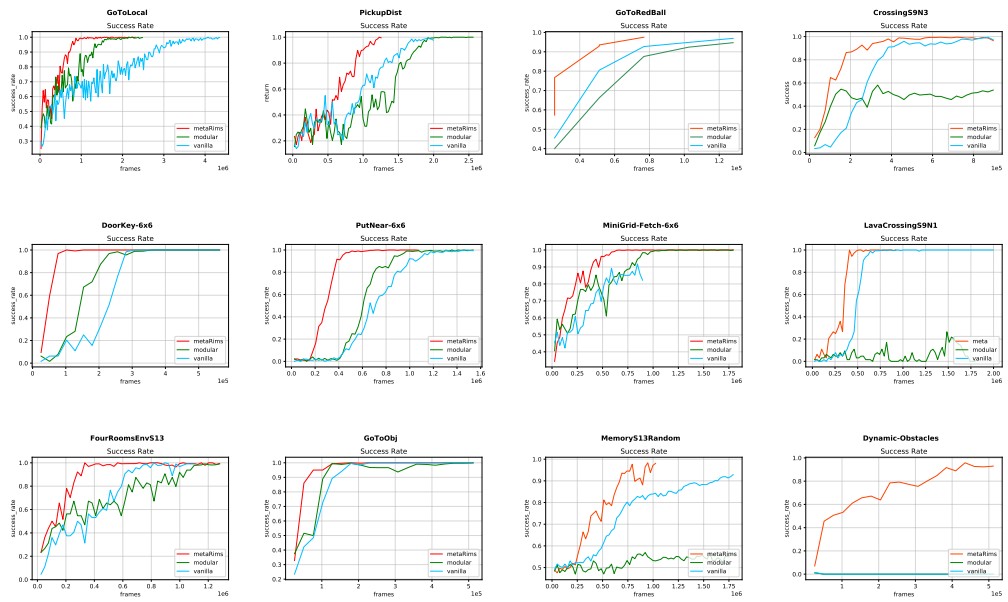

Figure 4: **Sample Efficiency:** The proposed method (labeled "metaRims") (red), outperforms the baselines of "modular" RIMs (green) and "vanilla" LSTM (blue) in terms of the Mean Success Rate

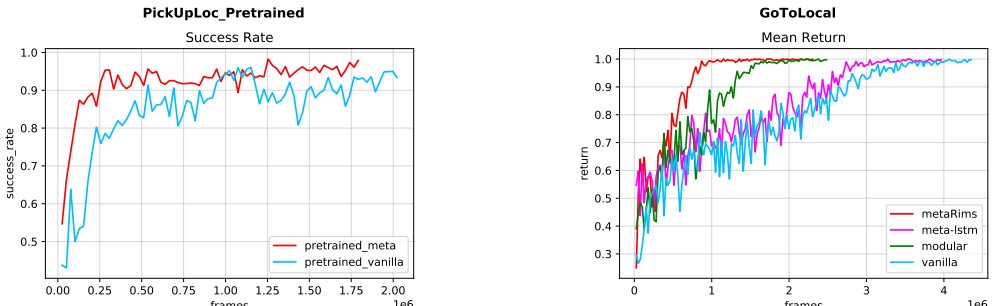

Figure 5: **Knowledge for Curriculum Learning and Ablation study experiments**: (a) The proposed model and the baseline are trained on an easier environment and then fine-tuned on a more difficult environment such that the environments share a subset of core competencies. The proposed method outperforms the baseline, indicating a better reuse of the previous knowledge. (b) To understand the importance of meta-learning, we trained a vanilla LSTM baseline in a similar meta-learning fashion, called "meta-lstm", which led to an improved the performance over vanilla setup.

the slow meta loop. We found an improvement in this "meta-LSTM" setup, see Fig. 5(b), showing positive evidence towards the benefits of meta-learning in the proposed approach.

***Sparsity in module activation and Slow-factor of Outer loop:*** In our experiments, we found that updating only a sparse subset of modules performs much better than updating all the modules in the fast learning loop, indicating that sparsity plays an important role. For the slow outer loop, we found that increasing the length of the interaction span helps in faster learning, see Fig. 6 (a).

***Value Function Visualization:*** We plot the value function predicted by the agent along different timesteps of a fixed length input sequence and plot the values across nine episodes of a meta episode, see Fig. 7. Teasing apart one of the episodes, we find that the value function shoots to a high value when the agent sees the target, and topples down as soon as the target is achieved, indicating that the agent is indeed finding good states and is acting according to a properly learnt value function.

***Visualizing Module Activations:*** We plotted module activations across two environments and found that the activations are quite diverse with no dead modules, indicating an active participation from all

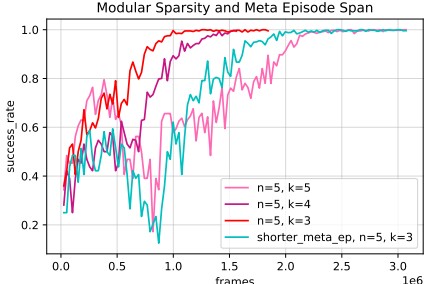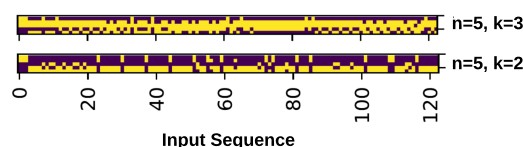

Figure 6: **Ablations:** (a) **Sparsity in Module Activations and Slow-Factor for outer loop**: As shown here, sparse activation and update of modules in the inner loop leads to faster learning and better performance. Also, longer spans of meta episodes in outer loop help in faster learning. (b) **Module Activations:** A plot of module activations ($y$-axis) for a fixed-length input sequence ($x$-axis) for two of the environments for settings ($n = 5, k = 3$) and ($n = 5, k = 2$) shows a diverse and active participation from all modules (no dead modules) to dynamically respond to the inputs received.

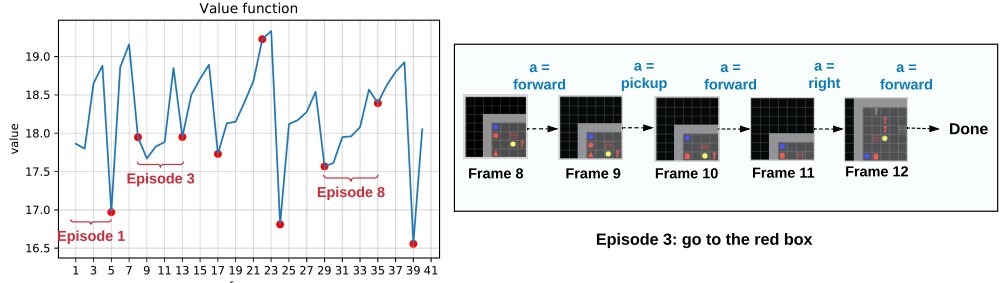

Figure 7: **Value function visualization**: The predicted value function plotted across nine episodes of a meta episode. Each red marker marks a new episode and "a" is the action taken. The frames shown are the partial observations received by the agent within one such episode. We can see that the agent understands potentially good states, the right targets and the states that need to be explored further.

of the modules to dynamically attend to variations in the observed input sequence, see Fig. 6 (b) and Appendix A.2 for further discussion.

***Reversing the Role of Attention Parameters:*** In order to evaluate the choice of using attention parameters as meta parameters in the proposed setup, we reversed the roles of the parameters such that the attention parameters act as quickly changing parameters in the inner loop and the parameters of the dynamic modules act as slowly changing. We found that this did not perform very well, see Fig. 8, emphasizing the particular importance of the role of attention parameters acting as slowly changing meta-parameters in the proposed setup.

***Importance of Fast and Slow Update Loops:*** We compared the proposed setup with a setting in which, instead of learning attention parameters slowly as meta-parameters in the outer loop, we reduced the learning rate of the attention parameters by one-fourth of the learning rate of other parameters. This way, all the parameters are learnt at the same time, but attention parameters are updated much more slowly. We found in this particular setting, the model does not perform very well, and the learning slows down significantly as compared to the proposed setup, see Fig. 8.

***What are the Modules Learning:*** We visualized the module activations of a trained agent for the MemoryS13Random environment across 20 episodes and tracked the agent's behavior towards the end of the episode when it is about to reach the goal , see Appendix Fig. 11 and Fig. 12. We found that (1) When the agent is about to reach the goal, module 5 is always activated, (2) Module 2 is activated when the agent is about to reach the junction at the end of the hallway indicating different roles these modules can play in different parts of the state space.

***Specific Roles of the Active Modules:*** To further validate the importance of the specific roles played by the active modules, we randomly turned off a subset of active modules. Specifically, we first randomly deactivate one module and then two of the three active modules. As expected, the performance degrades significantly, such that for 20 episodes, with 3 active modules, the agent takes 192 frames, while it takes much longer as modules are gradually deactivated, with 243 frames for two active and 2179 frames with only one active, see Fig. 9.

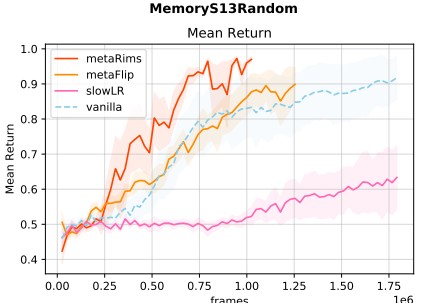
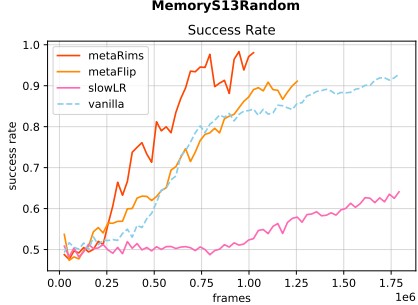

Figure 8: **Ablation: (1) Choice of attention parameters as slow-parameters:** In order to evaluate the choice of learning attention parameters as slow-parameters in the proposed setup, we experimented by reversing the roles such that the attention parameters are quickly changing parameters and the module parameters act as slow parameters. As seen from the curve labeled "metaFlip", this reversed-parameter setting does not do as well in terms of both Mean Reward and Success Rate. **(2) Slowing down the learning rate for Attention Parameters:** We also compare the proposed setup with a setting in which the learning rate of the attention parameters is reduced by one-fourth of the learning rate of other parameters and all parameters are learnt at the same time. The curve labeled "slowLR" shows that just lowering the learning rate of attention parameters does not help, and in fact slows down the learning, emphasizing the importance of the two learning loops in the proposed setup.

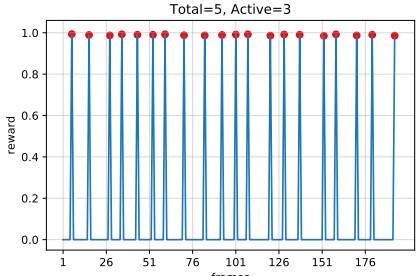
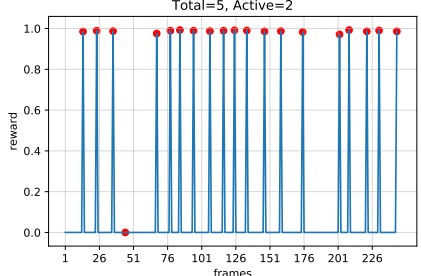

Figure 9: **Ablation: Special role of the winner modules** Turning off modules that win the competition significantly degrades performance, and agent takes much longer to complete an episode.

## 5 CONCLUSION

This paper investigates using a meta-learning approach on modular architectures with sparse communication (as in RIMs (Goyal et al., 2019)) to capture short-term vs long-term aspects of the underlying mechanisms in the data generation process, by considering parameters of attention mechanism as meta-parameters and parameters of the recurrent modules as parameters. Meta-learning is directed towards sample efficiency in the face of changes in distribution and tasks. The experimental results on grounded language learning tasks in the reinforcement learning setting strongly indicate that the combination of meta-learning combined with dynamically connected modular architectures with sparse communication, leads in many ways to superior results in terms of improved sample efficiency (faster convergence, higher mean return and success rates), and an improved transfer across tasks in a curriculum, both as zero-shot transfer and with adaptation. Ablation studies further confirm that using a meta-learning approach to update different parameters of the network over different timescales leads to improvements in sample efficiency as compared to training all the parameters at once. Overall, these results point towards an interesting way to perform meta-learning and attention-based modularization for better sample efficiency, out-of-distribution generalization and transfer learning. In the present work, different modules are still trained end-to-end to optimize a single objective, and future area of research would investigate how to train different modules, where each module has a separate objective and optimizing the module specific objective optimizes the global objective.

## 6 ACKNOWLEDGEMENTS

The authors would like to thank Stefan Bauer and Alex Lamb for useful discussions. This project internally at Mila was know as "Meta-Attention Networks".

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

# A  APPENDIX

## A.1  IMPLEMENTATION DETAILS

We used a variety of environments from MiniGrid and BabyAI (Chevalier-Boisvert et al., 2018) that provide a partial and egocentric view of the state of the environment to the agent. The reward is sparse and a positive reward is received only if the agent successfully reaches the goal. A penalty is awarded based on the number of steps taken to reach the goal, calculated as $1 - 0.9n/n_{max}$, where $n_{max}$ is the maximum number of steps allowed for a given environment and depends on the difficulty of the environment such that more difficult environments have a larger value of $n_{max}$. If the agent is not able to complete the task within $n_{max}$ steps, the episode ends and it gets a zero reward. The environments have an increasing level of difficulty in an systematically incremental manner. These settings of partial observability, sparse rewards and a systematic increase in the difficulty levels make the task for reinforcement learning algorithms sufficiently difficult.

The observation received by the agent consists of two parts: (1) an RGB image for the partial observation of the agent's field of view, and (2) a language instruction containing the mission statement that defines the task. The image part of the observation is processed through an encoder, and an embedding is computed for the textual mission statement, which are then fed to an ensemble of recurrent modules; see Fig. 1 for a full visualization of the architecture layout.

We used the Proximal Policy Optimization (Schulman et al., 2017) with parallelized data collection of rollouts collected by multiple parallel processes. For generalized advantage function, we used $\lambda = 0.99$, and discounted future rewards by a factor of $\gamma = 0.99$. Throughout the experiments, we present the mean-reward (R) and success-rate (S) of the agent, where the mean reward is the average reward across multiple runs, and the success rate represents the percentage of times the agent is able to successfully reach the goal within the $n_{max}$ timesteps. For all of our environments, we used $n = 5$ total modules, with only $k = 3$ of them active at any given time. Further details on the specifics of each environment are provided in the Section A.4.

## A.2  ABLATION: VISUALIZING MODULE ACTIVATIONS

In the proposed setup, the modules are dynamically selected and activated based on their relevance to the current input. In order to visualize the diversity and frequency of module activations, and to analyze the effect of environment dynamics on the activation patterns of the modules, we tracked the module activations of a trained agent along a fixed length input sequence obtained by tracking the agent navigating across the environment. We plotted these activated modules for two different environments - DynamicObstacles and PutNear, see Fig. 6 (b).

We find that for environment DynamicObstacles, in which the agent has to take more frequent decisions to figure out the best action due to quite engaging environment dynamics, such as multiple distractor objects and dynamically moving obstacles, the module activations are more diversely activated. In both the environments, different modules are active for different inputs, and all of them get activated at some point depicting an active engagement throughout the agent's interaction. However for the environment, PutNear, which has relatively fewer moving parts during the interaction, some modules are consistently more active than the others. However, all modules do get activated at some point, leaving no dead or always inactive modules.

This study has a direct analogy with how humans engage with their environments: a visual input with an enriched set of objects and more frequent interactions with the environment will need a higher engagement and continuous application of a wider set of skill sets. On the contrary, an environment which is relatively simple and has simpler input observations would need a lower engagement, and only a few components / skill-sets would mostly be able to handle and decide the best course of actions.

## A.3  ABLATION: VISUALIZING EPISODES AND MODULE ACTIVITY PATTERNS

In order to visualize the module activity pattern of an agent, and to understand the patterns in what the modules are learning, we plotted the module activity patterns over 20 episodes, see Fig. 11 and Fig. 12. On visually inspecting the behavior of the agent over these 20 episodes, we found that some

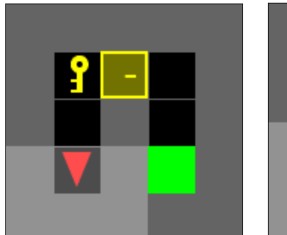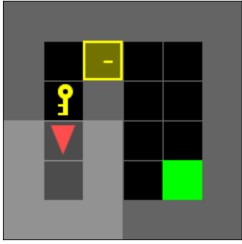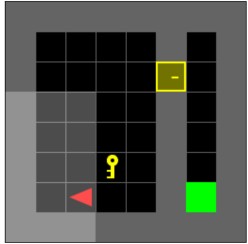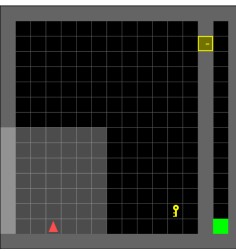

Figure 10: **Zero-shot transfer environments**: A series of DoorKey environments with an increasing level of difficulty from left to right; as the rooms get larger, the trajectories become longer and the rewards become sparser, making the tasks progressively more difficult. At the same time however, these environments share an underlying structure, making them suitable for evaluating policy generalization and transfer. We trained our agent in the leftmost (easiest) setting and evaluated it on the more difficult environments.

specific modules are always activated when a certain state space is reached: Module 5 is always activated when the agent is about to reach the matching object (the end goal), and Module 2 is always activated when the agent is around the second to the last step and when the agent is at the end of the hallway at the split. This shows a specific assignment of roles that the modules undertake to achieve an optimal behavior.

### A.4 ENVIRONMENTS: MINIGRID AND BABYAI

We trained our agents on several environments from MiniGrid and BabyAI (Chevalier-Boisvert et al., 2018), such as GoToLocal, PickupDist, Dynamic Obstacles, DoorKey, PutNear, Fetch, Four-RoomsS13, GoToObj, GoToRedBall, GoToRedBallGrey, MultiRoom, LavaCrossingS9N1 and Pick-UpLoc. In each of these environments, the observation consists of an RGB image of the agent's partial field of view, and an instruction textual string that contains the mission that the agent needs to solve. Fig. 1 visualizes the model architecture and the processing of the input observation. Further details on these environments, with the hyperparameters used for the recurrent modules, are provided below.

**Hyperparameters:** For all of the environments, we used a total of $n = 5$ modules with $k = 3$ active at any give time. More details on each environment are provided below.

**GoToLocal:** The agent is presented with a number of objects in a single room with no doors, and is asked to go to one of them as specified in the mission statement.

**PickupDist:** A single room, with no doors, has a number of objects and the agent needs to pick the object instructed in the mission statement.

**Dynamic Obstacles:** This environment contains moving obstacles in a single room, and the agent has to reach the goal located in a corner of the room while not colliding with these dynamically moving obstacles. A large penalty is subtracted if the agent collides with an obstacle. This environment is particularly useful to train moving robots in a dynamically moving objects setting.

**DoorKey:** In this environment, the agent needs to find a key in the current room, pick the key and unlock a door, and then reach the goal located in the other room.

**PutNear:** The mission statement specifies an object that has to be put near another object. In a room of multiple objects, the agent has to find the correct objects and put them in the right location as specified by the instruction statement, thus requiring both spatial and visual understanding of multiple objects.

**Fetch:**     The agent is presented with multiple objects of different colors and shapes, and it needs to find the object that is instructed to be fetched in the mission instruction string. A negative reward is given if a wrong object is picked.

**MemoryS13Random:**     The agent starts in a room in which it sees an object, and has to remember this object when it reaches the end of the hallway which ends in a split. One of the sides of this split randomly contains the same object as it saw in the beginning and it has to choose that matching object.

**FourRoomsS13:**     This is a multi-room environment, consisting of four rooms, connected by four gaps in the walls, such that the goal is randomly placed in one of the rooms. The location of the agent is also random when the episode starts.

**GoToObj:**     This is a single room environment with a single object, specified in the mission string, that the agent needs to reach to.

**GoToRedBall:**     The goal is to reach a red ball in a room containing distractors and other obstacles.

**GoToRedBallGrey:**     The room consists of multiple grey distractors and the agent needs to reach the red ball, without requiring any unblocking.

**MultiRoom:**     The environment consists of a series of interconnected rooms, and the agent has to unlock the door to navigate to the goal in the next room.

**LavaCrossingS9N1:**     The environment consists of horizontal and vertical strips of lava running across the room, and the agent has to reach the goal while avoiding them. If the agent touches the lava, it dies and the episode ends with no reward.

**PickUpLoc:**     A single room environment in which the goal is to pick up an object described by its location.

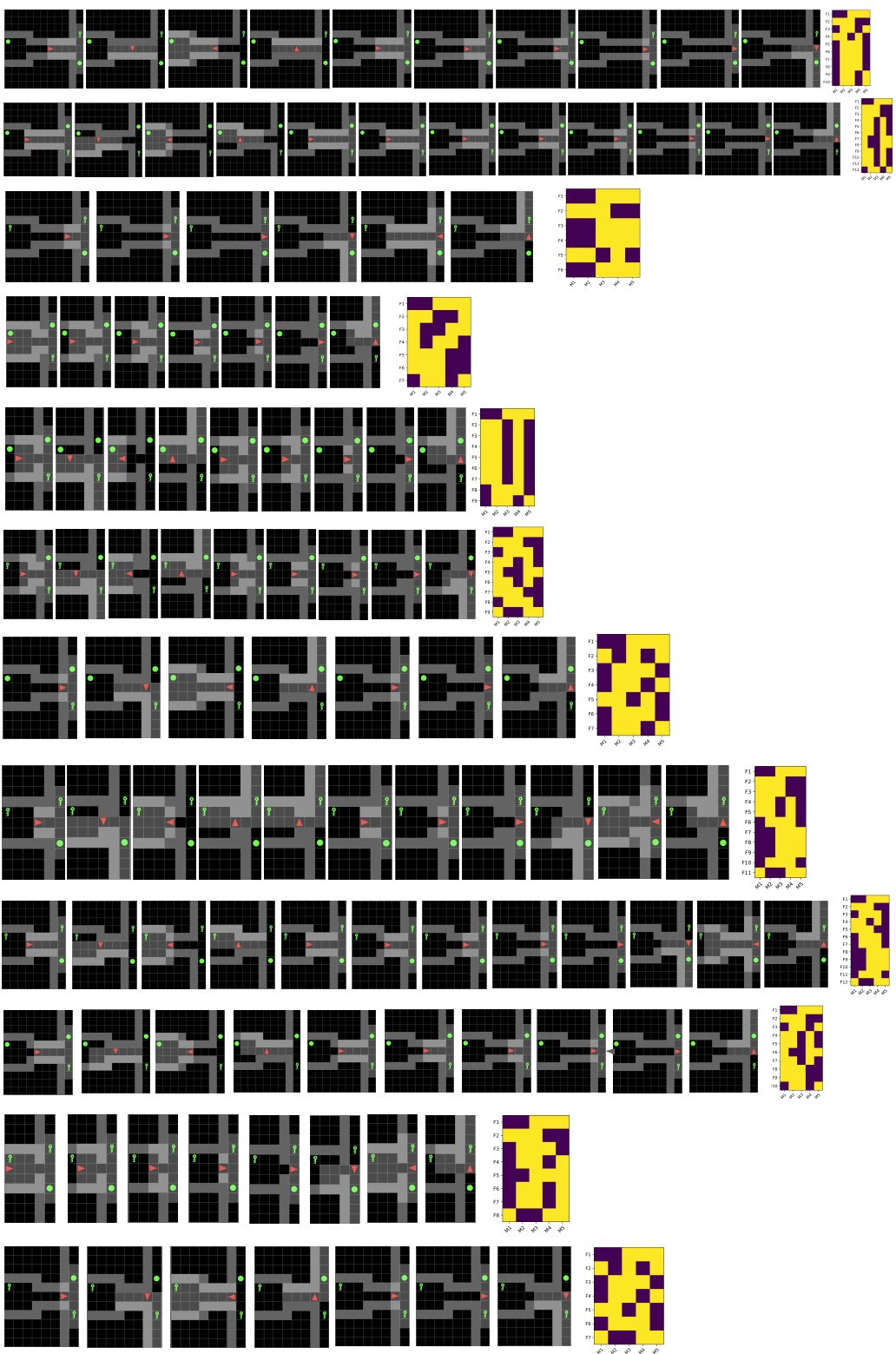

Figure 11: **Visualizing Activity of Modules as the Agent Navigates**: Visualizing Episodes 1-12, along with the activations of 5 modules such that 3 are active at any give time. We find that: (1) When the agent is about to reach the goal, Module 5 is always activated, and (2) Module 2 is always activated when the agent reaches the split junction at the end of the hallway towards the last few steps of the episode, indicating a particular trend in the module activations in certain parts of the state space.

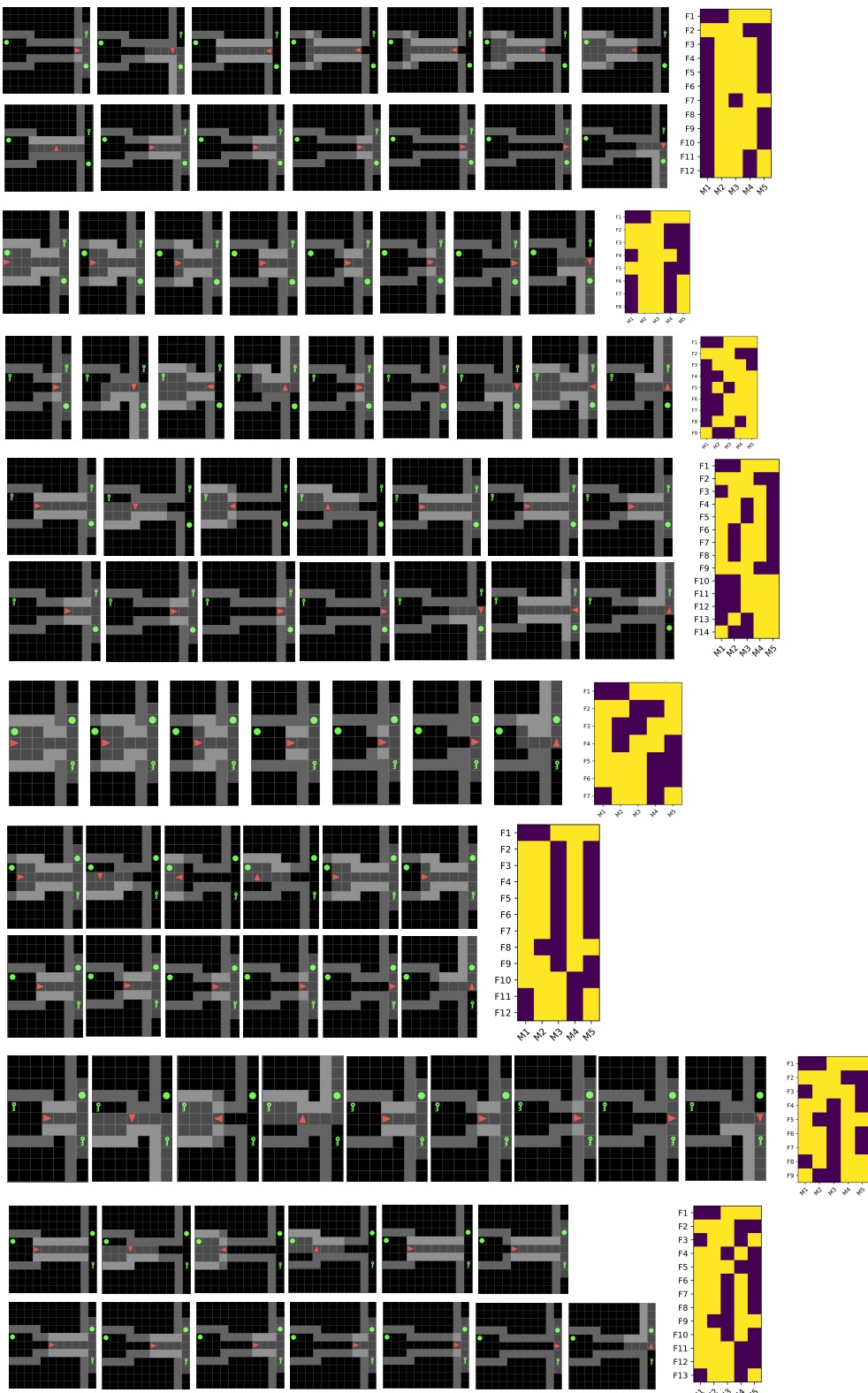

Figure 12: **Visualizing Activity of Modules as the Agent Navigates**: Visualizing Episodes 13-20, along with the activations of 5 modules such that 3 are active at any give time. We find that: (1) When the agent is about to reach the goal, Module 5 is always activated, and (2) Module 2 is always activated when the agent reaches the split junction at the end of the hallway towards the last few steps of the episode, indicating a particular trend in the module activations in certain parts of the state space.