# OpenReview forum: "Fast And Slow Learning Of Recurrent Independent Mechanisms"
_ICLR.cc/2021/Conference — ICLR 2021 Poster_

### Official Review · AnonReviewer3 · 2020-10-28
**Concerns about motivation.**

**Rating:** 5
**Confidence:** 3

**Review:**

This paper proposes a meta-learning (called) method for the model based on RIMs architecture to realize fast adaptation on new unseen tasks.

The base model is the same as RIM, which is composed of multiple independent LSTM modules. An input attention is used to determine which independent modules are activated, taking output state from them. Another attention fuses information from activated modules. This paper proposes to update the parameters of attention less frequently than LSTM modules instead of updating them with the same frequency, as they believe that attention should capture a more stable distribution of the task.

However, there are some concerns about the motivation of this paper. There is no doubt that meta-learning can achieve faster adaptation between domains. But why should we divide the parameters into two sets? How about treating all parameters as a single set and updating them under a traditional meta-learning way in which parameters are separately optimized on each distribution and the average loss among distributions is used for gradient calculation. In addition, will the distribution change much for different frames within the same task?

This paper is a bit hard to understand as there are too many long sentences with complex clauses. It is better be improved via revision.

Some questions:

1. concerns about motivation.

2. the current strategy is similar to setting a lower learning rate for parameters of attention modules. Does the paper have any results to prove that the proposed method is better than using different learning rates?

3. why setting the step length of slow update fixed as four times of fast update? Would it be better to determine it dynamically? (e.g. if the frame distribution difference has exceeded a threshold then taking outer update)

4. it is unclear about the parameter updating strategy of baselines (vanilla LSTM and RIM). All parameters in them will be updated under the same frequency. But is the frequency the same as the fast update or slow update in the method.

---

> ### Author Response · Authors · 2020-11-13
> **Concerns of the motivation and Updated the manuscript: Part (1/2)**
>
> We thank the reviewer for the positive and constructive feedback. We have conducted more experiments in order to answer your questions. We thank the reviewer as we believe these extra experiments have improved the presentation of the paper.
>
> > "concerns about motivation"
>
> We thank the reviewer  for bringing up this question.
>
> -  The goal of the paper is to come up with architectures and learning rules, which can help in improving the generalization performance of the model.
> - A learning agent interacting with its environment is likely to be faced with situations requiring novel combinations of existing pieces of knowledge.
> - We hypothesize that such a decomposition of knowledge is particularly relevant for being able to generalize in a systematic way to out-of-distribution changes [1,2,3,4].
> - To study these ideas, we propose a particular training framework in which we assume that the pieces of knowledge an agent needs, as well as its reward function are stationary and can be reused across tasks.
> - We find that a modular architecture combined with learning different parts of the model on different time-scales can help in decomposing knowledge into reusable pieces such that the resulting model is not only sample efficient but also generalizes well across changes in distribution.
>
> [1] Generalization without systematicity https://arxiv.org/abs/1711.00350
> [2] Systematic generalization what is learned, and can it be learned https://arxiv.org/abs/1811.12889
> [3] Meta transfer objective for learning to disentangle causal mechanisms https://arxiv.org/abs/1901.10912
> [4] Recurrent Independent Mechanisms.  https://arxiv.org/abs/1909.10893
>
> > "But why should we divide the parameters into two sets? How about treating all parameters as a single set and updating them under a traditional meta-learning way in which parameters are separately optimized on each distribution and the average loss among distributions is used for gradient calculation."
>
> One of the primary goals of the proposed work is to decompose the knowledge into independent pieces such that they can be reused and combined dynamically to generalize to out of distribution samples, and that is why we considered a modular architecture.In the previous work, modular architectures [1,2,3,4] have shown to perform better on out of distribution tasks.
> Meta-learning over a set of distributions can be interpreted as learning different types of parameters corresponding to short-term vs long-term aspects of the mechanisms underlying the generation of data. These are respectively captured by quickly-changing parameters and slowly-changing meta-parameters. A learning agent interacting with its environment is likely to be faced with situations requiring novel combinations of existing pieces of knowledge, and hence the agent needs to adapt quickly when there’s a change in distribution. We can make parallel connections to meta-learning, where the inner loop can be considered as fast adaptation to the distribution change (by changing the parameters of a subset of modules) and outer loop can be considered as learning the stationary meta-parameters of the model.
>
> Recently [a, b] used meta-learning to learn causal mechanisms or causal dependencies between a set of high-level variables, which inspired the approach presented here.  The 'modules' in their work are the conditional distributions for each variable in a directed causal graphical model. The inner-loop of meta-learning also allows the modules to be adapted within an episode (corresponding to an intervention distribution), while the outer-loop of meta-learning discovers how the modules are connected (statically) to each other to form the graph structure of the graphical model.
> - [a] Meta transfer objective for learning to disentangle causal mechanisms https://arxiv.org/abs/1901.10912
> - [b] Towards Neural Causal models with unknown Interventions, https://arxiv.org/abs/1910.01075

---

> > ### Author Response · Authors · 2020-11-13
> > **Different Learning rates: Part (2/2)**
> >
> > >  "The current strategy is similar to setting a lower learning rate for parameters of attention modules. Does the paper have any results to prove that the proposed method is better than using different learning rates?"
> >
> > This is an interesting setting, and we would like to add a bit more context on how our setting is different from having a lower learning rate for the attention parameters. The two differ as follows:
> >
> > In the proposed setup, not only are the different parameters learnt at different speeds, but also different parameters look at different time horizons. The parameters in the inner loop tend to capture the quickly changing aspects of the environment over much shorter timespans. While, in the slower loop, the attention parameters get to see much longer sequences and are updated much less slowly, enabling them to capture more stable patterns by looking over relatively longer time spans.
> >
> > In order to better evaluate the setup proposed in the paper with the setting of having a slower learning rate for the attention parameters, we ran more experiments  on the MemoryS13Random environment by reducing the learning rate of attention parameters to one-fourth of the learning rate of other parameters (to imitate the fast and slow learning in the current setup) while learning all the parameters at the same time; we refer to this setup as “slowLR”. We found that the “slowLR” baseline, is less sample efficient as compared to the  proposed method (see Fig. 8, in the main paper), hence showing the benefits of learning different parameters over different timescales.
> >
> >
> >
> > > "Why setting the step length of slow update fixed as four times of fast update? Would it be better to determine it dynamically? (e.g. if the frame distribution difference has exceeded a threshold then taking outer update)"
> >
> > This is a very interesting proposal, and we agree that having such a dynamic setting might provide more flexibility to the network by letting it adjust the slowness-factor based on the difficulty of the environment. In the current work, in one of our ablation studies, we found positive evidence towards benefits from slowing down the outer loop, see Fig 6a, curve labeled “shorter_meta_ep, n=5, k=3”, in which “four times longer than fast update” does better than “two times longer the fast update”. We, however, do not see any specific limitations in converting this to a dynamic setting, and we would consider this as an extension of the current framework in the future work.
> >
> >
> >
> > > "it is unclear about the parameter updating strategy of baselines (vanilla LSTM and RIM). All parameters in them will be updated under the same frequency. But is the frequency the same as the fast update or slow update in the method."
> >
> > Yes, we keep the frequency of updates consistent across the baselines and the proposed setup. We will make it more clear in the paper.

---

> > > ### Author Response · Authors · 2020-11-17
> > > **Anything else you would like us to respond to ?**
> > >
> > > Dear Reviewer,
> > >
> > > We thank the reviewer for their feedback and valuable comments.
> > >
> > > Since the first phase of response period is closing soon, if you have time and could indicate if there are any other concerns of yours which we have not addressed, we'd be happy to take a look.
> > >
> > > Thanks for your time.

---

> > > > ### Author Response · Authors · 2020-11-20
> > > > **Studying behaviour of different modules.**
> > > >
> > > > Hello,
> > > >
> > > > We ran more experiments to directly address your concerns about the motivation, and what different modules are doing.
> > > >
> > > > Env Setup: We take a trained agent on the MemoryS13Random Environment from the MiniGrid, which is a particularly interesting environment in which the agent gets to see an object in the beginning of the episode. The agent then has to navigate through a narrow hallway and has to match the object initially shown with the correct object at the end of the hallway which splits into two paths. An efficient strategy to navigate in this environment is to get the object shown in the beginning in the field of view and match it with the right object by navigating to the end of the hallway split.
> > > >
> > > > In order to analyze the behavior of the modules, we analyze the activity patterns of all the five modules over 20 episodes, along with the incoming visual observations, (please refer to Appendix Fig. 11 and Fig 12, and with an ablation discussion under subsection: “What are the Modules Learning” (Page 8) in Section 4.4 in the main paper). We track the agent’s behavior towards the end of the episode when the agent is about to reach the goal. Specifically, we found that module 5 always becomes active when the agent is about to reach the goal towards the end of the episode, and module 2 gets activated when the agent reaches the junction point (i.e., end of the hallway split and around the last 2-3 steps of the episode). We note that this behavior is consistent across all episodes, showing a particular trend in the module activations in certain parts of the state space.
> > > >
> > > > Let us know if there's something we can do to address your concerns and increase your score.

---

### Official Review · AnonReviewer2 · 2020-10-28
**Interesting Extension on Prior Art, and Compelling Results**

**Rating:** 7
**Confidence:** 5

**Review:**

**Objective of paper**: Exploring methods that exhibit better systemic out-of-distribution generalization/adaptation properties. Specifically, exploring how modularity, combined with learning at different time scales, can help with this.


**Central claim**: By expanding Recurrent Independent Mechanisms with a meta-learning layer to facilitate learning at different time scales, it is possible to achieve better within-distribution and out-of-distribution/systemic generalization and fast adaptation to new tasks.


**Strong points**:
1.	Important research direction: I believe this research direction has a lot of value in exploring and improving the way our current deep learning systems can exhibit better systemic generalization.
2.	Novelty: While most of the individual components presented in the paper are not novel, the particular combination of these components and the empirical evaluation seem novel to me. (That being said, the idea of meta-learning gating mechanisms to combat catastrophic forgetting and interference (arguably the central contribution of this paper) has been explored in continual learning before – for example [1]).
3.	Compelling experimental results: In terms of raw performance, the proposed method clearly outperforms its Recurrent Independent Mechanisms (RIM) and LSTM counterparts in sample complexity, better systemic generalization and faster adaptation. I also think that the tasks picked for evaluation are suitable to investigate systemic generalization.


**Weak points**:
1.	Lack of architectural/algorithmic benchmarks in experiments: Given that systemic generalization, as well as combatting catastrophic forgetting, are active research areas, it would have been nice if the paper compared its results with alternative, competing approaches. If you think there are no other comparable approaches in the literature, then it would be nice to discuss why in the related works section. It might also be helpful to try out simpler versions of the proposed method to see what component really made the approach work. For example, to really double down on how much the meta learning component matters, could the authors try training all of the parameters at the same time (i.e. no meta-learning step) but with a small learning rate assigned for the attention parameters (maybe ¼ of the current learning rate, since you update it 4 times less than the dynamics parameters, if I’m not mistaken )?
2.	Claim about knowledge transfer not supported well enough: Besides the fact that the knowledge transfer results are reported on a small percentage of the environments, I’m not sure if we can conclude “knowledge transfer” is responsible for better performance when one uses models pretrained on similar tasks. Couldn’t this be due to some simpler explanation, such as better optimization landscape for pretrained models? I think the authors can address this by first pretraining the models on completely irrelevant tasks where knowledge transfer is not expected to help and showing that in this case the reported gains in performance are *not* shown. It might also help elaborate on what is meant by “knowledge”.
3.	Hyperparameter tuning procedure not described: The hyperparameter tuning procedure was not described in the paper (I might have missed it too?). It would be particularly interesting to read more about how the authors arrived at updating the meta parameters once for every 4 inner loop updates.
4.	Lack of “hard” environments: It seems from the results that both LSTM and RIM already perform well on most of the tasks shown in the paper (though slower and with more data). Is this a correct assessment? Are there tasks that only the proposed methods can solve? (I guess zero-shot learning results are somewhat of an example, but it would be nice to see harder environments too)
5.	(minor) The fact that number of modules (N) and number of active modules allowed at a time (k) are fixed: I guess this can be viewed as a weakness, thought this is shared in RIM as well, and not that relevant to what is being evaluated/investigated in the paper.


**Decision**: This paper makes an algorithmic contribution to the systemic generalization literature, and many in the NeurIPS community who are interested in this literature would benefit from having this paper accepted to the conference. I'm in favour of acceptance.


**Questions to authors**:
1.	Is there a specific reason why you reported results on knowledge transfer (i.e. section 4.3) only on a few select environments?
2.	As mentioned in the “weak points” section, it would be nice if you could elaborate on
3.	Is it possible that the caption of Figure 4 is misplaced? That figure is referenced in Section 4.1 (Improved Sample Efficiency), but the caption suggests it has something to do with better knowledge transfer.
4.	If you have the resources, I would be very interested to see how the “small learning rate for attention parameters” benchmark (described above) would compare with the proposed approach.
5.	In Section 4, 1st paragraph, you write “Do the ingredients of the proposed method lead to […] a better curriculum learning regime[…]”. Could you elaborate on what you mean by this?


[1] Beaulieu, Shawn, et al. "Learning to continually learn." arXiv preprint arXiv:2002.09571 (2020).

---

> ### Author Response · Authors · 2020-11-15
> **Ran more experiments to answer your question: Learning rate, and Knowledge Transfer (Part 1/2)**
>
> We are enthused that the reviewer find the proposed research direction "valuable" and experimental results "compelling".
>
> > "For example, to really double down on how much the meta learning component matters, could the authors try training all of the parameters at the same time (i.e. no meta-learning step) but with a small learning rate assigned for the attention parameters (maybe ¼ of the current learning rate, since you update it 4 times less than the dynamics parameters, if I’m not mistaken )?"
>
>
> Thank you for bringing up this interesting insight. Our setting is actually quite different than just having a different learning rate for attention parameters as: in the proposed meta-learning setting, not only do different parameters learn at different speeds, but also, they learn on different timespans to better capture the quickly changing vs slowly changing aspects of the input sequence.
>
> In order to quantitatively analyze this hypothesis, we ran some more ablation studies as suggested by the reviewer. Essentially, we trained a model such that the learning rate of the attention parameters is reduced to one-fourth of the learning rate of the other parameters of the network, and we refer to this setting as “slowLR”. We found that the “slowLR” method does not perform as well as the proposed method and is significantly less sample efficient. ( see Fig. 8 in the main paper).
>
> We also found that all of the other ingredients of the proposed setup, i.e. modular architecture, meta learning setup, sparse activation of modules, and slower learning of attention parameters over different timescales, play an important role towards the performance gains, please see Fig. 5b, Fig. 6a, and Fig 8 for some ablation studies and discussion on these.
>
> > "Claim about knowledge transfer not supported well enough"
>
> We ran more experiments to answer this question. We first describe the training setup.
>
> Pre-Training Env: Lava Crossing Env. The agent has to reach the green goal square at the opposite corner of the room, and must pass through a narrow gap in a vertical strip of deadly lava. Touching the lava terminates the episode with a zero reward. This environment is useful for studying safety and safe exploration.
>
> Transfer Env: (Dynamic Obstacle Env) This environment is an empty room with moving obstacles. The goal of the agent is to reach the green goal square without colliding with any obstacle. A large penalty is subtracted if the agent collides with an obstacle and the episode finishes. This environment is useful to test Dynamic Obstacle Avoidance for mobile robots with Reinforcement Learning in Partial Observability.
>
> - For LSTM baseline, directly training on Dynamic Obstacle Env for 20M time-steps achieves a 76% success rate.
> - For modular baseline, directly training on Dynamic Obstacle Env for 20M time-steps achieves a 81% success rate.
> - For LSTM baseline, pre-training on Lava Crossing, and transfer on Dynamic Obstacle achieves a 71% success rate.
> - For modular baseline, pre-training on Lava Crossing and transfer on Dynamic Obstacle Env  achieves a 79% success rate.
>
> We note that there's not much shared structure b/w the two env. In this env, the LSTM baseline after training on the pre-training env and then transfer (or fine-tuning) on the Lava Crossing env PERFORMS slightly worse as compared to directly training on the LAVA Crossing env. Similarly, one can find that the modular baseline, also performs slightly worse (but better than LSTM)  as compared to directly training on Dynamic Obstacle env. This hints that the knowledge transfer is because of the shared underlying structure as compared to other reasons.
>
> > "Hyperparameter tuning procedure not described: The hyperparameter tuning procedure was not described in the paper (I might have missed it too?). "
>
> In order to not have to tune the hyperparameters for each environment and to test the models in their most general form, we do not tweak the hyperparameters for the total number of modules and the top-k modules and rather keep it consistent across all the environments. The length of the outer loop was also not tweaked for different environments.
> In our ablation studies, we found improvements from slowing down the outer loop, see Fig. 6a, in which the curve labeled “shorter_meta_ep, n=5, k=3”, is for the setting when the outer loop is slowed down by two times, as opposed to four times in the current setting, and slowing down four times slower setting performed better. Note that however, both these setups do better than the baselines, showing the contribution of the meta-learning setup and learning at different timescales. That said, we don’t see any limitations in converting the slowness to a dynamic setting in the future work. We hope this helps in providing a better context on the thought process for the hyperparameters.

---

> > ### Author Response · Authors · 2020-11-15
> > **Other difficulties like exploration in "harder" environments. (Part 2/2)**
> >
> > > "Lack of “hard” environments: It seems from the results that both LSTM and RIM already perform well on most of the tasks shown in the paper (though slower and with more data). Is this a correct assessment? Are there tasks that only the proposed methods can solve? (I guess zero-shot learning results are somewhat of an example, but it would be nice to see harder environments too)"
> >
> > This is an interesting observation, and the main reason that we have the baselines mostly achieving the tasks is because we wanted to provide results from those environments which every method could solve to have a one-to-one comparison. For some more difficult environments, it might be possible that some of the methods would not converge. For example, we noticed such a thing with the Dynamic Obstacles environment.
> >
> >
> > > "The fact that number of modules (N) and number of active modules allowed at a time (k) are fixed: I guess this can be viewed as a weakness, thought this is shared in RIM as well, and not that relevant to what is being evaluated/investigated in the paper."
> >
> > We appreciate bringing up this concern. Although as already stated in the question, this is a feature of the RIMs and we do not explicitly evaluate it here. However, in our ablation studies, we have noticed that sparsity in module activations helps in better learning, see Fig 6a, and this is also something which has been also been found by the authors of the RIMs paper.

---

> > > ### Comment · AnonReviewer2 · 2020-11-18
> > > **Thank you for your response.**
> > >
> > > Thank you for your response. Especially the additional learning rate tuning and knowledge transfer experiments help increase the confidence in the proposed method and claims made in the paper. (I still think that including results from more difficult environments in the main body of the paper (such as results on Dynamic Environments) could further improve the paper and strengthen the claims if the proposed methods is the one that actually outperforms the competitors asymptotically.)
> > >
> > > Additional nitpick comment: I agree with some of the other reviewers that the description of how exactly the proposed algorithm works could be improved. In addition to Figure 2, perhaps an algorithm box might help clear up any remaining confusion.
> > >
> > > I’ll keep my score as is (Good paper, accept), and increase my confidence level.

---

> > > > ### Author Response · Authors · 2020-11-20
> > > > **More Results on Dynamic Environments**
> > > >
> > > > Hello,
> > > >
> > > > We thank the reviewer for increasing their confidence.  We are excited that our additional analysis and discussion helped in addressing your concerns.
> > > >
> > > > To address your concerns we have ran additional experiments with three more environments - *CrossingS9N3, LavaCrossing, and Dynamic Obstacles* (please refer to Fig. 3 and Fig 4, in the main paper)..
> > > >
> > > > In short, the proposed method outperforms all  the other baselines on these environments.
> > > >
> > > > - For CrossingS9N3, the proposed method is almost twice as sample efficient as compared to the vanilla lstm baseline, while RIMs perform worse as compared to LSTM baseline.
> > > >
> > > > - For LavaCrossing, the proposed method takes 30% less samples as compared to other baselines (LSTM as well as modular baseline).
> > > >
> > > > For Dynamic Obstacles, only the proposed method is able to achieve a high success rate as compared to other baselines.
> > > >
> > > >  We hope these additional experiments will help to address concerns of the reviewer.
> > > >
> > > > Let us know if there's anything we can do to increase your score. We are willing to spend time and effort in improving the paper.
> > > >
> > > > Thanks for your time.

---

### Official Review · AnonReviewer1 · 2020-10-28
**A decent paper with a great goal where several small issues outweigh the benefits**

**Rating:** 7
**Confidence:** 3

**Review:**

Summary of the paper:

The paper introduces a meta-learning approach to recurrent independent modules, with the goal to make RIMs adapt better to changing distributions, increasing generalization, and increasing sample efficiency. It postulates that RIMs, with their decomposition of independent information, can more easily adapt in these cases. The introduces approach wraps a "meta-learner" in form of two different update speeds around RIMs. The "inner" loop updates the RIM module parameters with large (fast) updates, while the "outer" loop, consisting of input and communication attention, does small (slow) updates. The authors then evaluate the approach on the reinforcement learning domains of "MiniGrid" and "BabyAI". Comparing their approach with an LSTM wrapped in a similar "meta-learning" layer, they show that their approach outperforms the LSTM, and is generally able to quickly adapt to changes in the environment.

Commentary on the goal of the paper:

Learning in non-stationary environements is an open area of research, with a lot of potential. Combining the intuitions between two different approaches - the modularity of a RIM, and the adaptivity of a "meta-learner" is a very appealing idea.

Strengths:
- The paper's main approach is both intuitive and simple
- The results are good, showing clear improvements over the baseline
- The paper is generally well-written, and well-structured

Weaknesses:
- The paper lacks detail in the description of the proposed approach. How exactly does the overall architecture work? How do the different attention mechanisms interact? From the explanations and descriptions given, I would not be able to reproduce the architecture.
- The paper should have focused more to outline the reasoning behind the fast-slow learning design: why does the "outer loop" learn slowly, while the "inner loop" learns slowly? This is against my intuitions: I would have expected the outer loop to learn fast, and the inner slow - after all, the appeal of modular architectures is to re-use the modules, not the composition.
- The paper overstates the novelty of their approach. Under "modular networks" on page 5, it claims that the approach of modular networks is "hampered by the amount of domain knowledge required to decide or predict (Johnson et al., 2017) how the modules should be parameterized, and how they should be connected with each other". While this is true for the early work of Andreas, this statement completely ignores later research that tries to address exactly this shortcoming ("Routing Networks" by Rosenbaum et al 2017; "Modular Networks" by Kirsch et al 2018; "Automatically Composing Representation Transformations as a Means for Generalization" by Chang et al 2018; "Recursive Routing Networks" by Cases et al, 2019). In particular "Routing Networks and the Challenges ..." by Rosenbaum, 2019 already mentions slowing the training of the outer loop.
- The paper falls short on qualitative analysis: what do the modules learn, what is transfered, and how does this help with non-stationary environments?

---

> ### Author Response · Authors · 2020-11-13
> **Description of the architecture and Cited Relevant Work: Part (1/2)**
>
> We are enthused that the reviewer find the paper both well written and well structured. We have conducted more experiments in order to answer your questions. We thank the reviewer as we believe these extra experiments has improved the presentation of the paper.
>
> > "The paper lacks detail in the description of the proposed approach. How exactly does the overall architecture work?"
>
>  The main flow of the approach is depicted in Fig. 1 and Fig. 2 of the paper:
>
> - The proposed  architecture consists of an ensemble of recurrent modules, and each module has its own independent parameters.
> - Given an input, only a subset of modules get activated and get to update their internal dynamics. This selection of relevant modules is facilitated by a soft key-value based attention mechanism. Each module only attends that particular information which is relevant for that module. This kind of “top-down” attention has seemed to improve generalization in previous work [a, b, c, d].
> - Different modules share information via a bottleneck of attention (communication attention).
> - The output from these recurrent modules is then fed to two separate policy and value function heads, which learn the policy and the value function respectively.
>
>
>
> - [a] Recurrent Independent Mechanisms. https://arxiv.org/abs/1909.10893
> - [b] Learning to combine Top-Down and Bottom-up signals in RNN  with attention over modules. https://arxiv.org/abs/2006.16981
> - [c] Top Down Control of visual attention, https://www.ncbi.nlm.nih.gov/pmc/articles/PMC2901796/
> - [d] Top Down Control of Visual Attention: A Rational Account.  https://papers.nips.cc/paper/2875-top-down-control-of-visual-attention-a-rational-account
>
> > “How do the different attention mechanisms interact?”
>
> There are two separate sets of attention mechanisms which operate as below:
>
> **Input attention**:  At every timestep, different modules make a query and keys/values are a function of the input. Different modules compete with each other such that a subset of the modules are activated at every time-step based on their relevance to the input.
>
> **Communication Attention**: This attention mechanism enables a sparse sharing of information between modules. An activated module can query other modules, while keys and values are a function of the hidden states of these modules.
>
> The full network is learnt in an end-to-end manner using a meta-learning setup such that the inner loop updates the parameters of a sparse subset of  modules in a fast manner, and the outer loop updates the attention parameters slowly over relatively longer time spans to capture more stable properties of the input sequence.
>
> > "The paper falls short on qualitative analysis: what do the modules learn, what is transferred, and how does this help with non-stationary environments?
>
> Thank you for bringing this interesting question. In order to understand a bit more on what the modules are learning, as well as variations across module activations, we performed some ablations studies in Section 4.4. We visualize the value function along with the incoming input images. We also look at the diversity of the module activations. Through these, we found that the agent is appropriately capturing the sub-goals, as depicted in the value function and input image frames shown in Fig 7. Also, a diverse set of module activations and no dead modules, also show how the modules are actively engaging to respond to the input variations, please refer to Fig. 6 b. This diversity is more evident for more diversified environments which also have a more diversity in their input. That said, we would be very happy to do more analysis .

---

> > ### Author Response · Authors · 2020-11-13
> > **Part (2/2)**
> >
> > > "The paper should have focused more to outline the reasoning behind the fast-slow learning design: why does the "outer loop" learn slowly, while the "inner loop" learns slowly?"
> >
> > Thank you for bringing up this hypothesis. We conduct more experiments to answer this question. We show that (1) swapping the role of parameters (in the modules) and meta-parameters (in the attention mechanism) does not work as well, (see Fig. 8) (2) only using meta-learning but not the RIMs modular architecture (i.e using LSTMs) does not work as well, and that (3) using only a modular architecture (i.e RIMs) but no meta-learning does not work as well as the proposed method (Fig. 5b).
> > Meta-learning over a set of distributions can be interpreted as learning different types of parameters corresponding to short-term vs long-term aspects of the mechanisms underlying the generation of data. These are respectively captured by quickly-changing parameters and slowly-changing meta-parameters. A learning agent interacting with its environment is likely to be faced with situations requiring novel combinations of existing pieces of knowledge, and hence the agent needs to adapt quickly when there’s a change in distribution. We can make parallel connections to meta-learning, where the inner loop can be considered as fast adaptation to the distribution change (by changing the parameters of a subset of modules) and outer loop can be considered as learning the stationary meta-parameters of the model.
> >
> > Recently [a, b] used meta-learning to learn causal mechanisms or causal dependencies between a set of high-level variables, which inspired the approach presented here.  The 'modules' in their work are the conditional distributions for each variable in a directed causal graphical model. The inner-loop of meta-learning also allows the modules to be adapted within an episode (corresponding to an intervention distribution), while the outer-loop of meta-learning discovers how the modules are connected (statically) to each other to form the graph structure of the graphical model.
> > - [a] Meta transfer objective for learning to disentangle causal mechanisms https://arxiv.org/abs/1901.10912
> > - [b] Towards Neural Causal models with unknown Interventions, https://arxiv.org/abs/1910.01075
> >
> > > "The paper overstates the novelty of their approach."
> >
> >
> > We thank the reviewer for pointing these references. We have updated the paper and referenced them accordingly. We note that these networks have an architecture which is composed  dynamically from several neural modules, where each module is meant to perform a distinct function. In such architecture different modules are applied one at a time, in contrast to the proposed architecture, where different modules can update their state and be used for prediction in **parallel**. The focus of this work is to automatically decompose knowledge into a set of independent modules as well as learn input-dependent connectivity between these modules. We show that using a meta-learning setup helps in learning better decomposition of knowledge such that the resulting model can generalize better out-of-distribution scenarios. We are not aware of any work where such networks which reviewer pointed to are used in the context of meta-learning.

---

> > > ### Author Response · Authors · 2020-11-18
> > > **Anything else you would like us to respond to ?**
> > >
> > > Hello,
> > >
> > > We thank the reviewer for their feedback and valuable comments.
> > >
> > > Since the first phase of response period is over, if you have time and could indicate if there are any other concerns of yours which we have not addressed, we'd be happy to take a look. We're very happy to invest the time to improve the paper further.
> > >
> > > Thanks for your time.

---

> > > > ### Comment · AnonReviewer1 · 2020-11-18
> > > > **A great rebuttal**
> > > >
> > > > The authors gave a great rebuttal of most of my concerns, and even updated the paper in time to show their effort. I will increase my score.
> > > >
> > > > However, the paper still falls short in answering what the modules learn, and why. I admit that this is a very difficult question to answer, if only because it is so difficult to analyze and visualize the training behavior of neural networks. Had the authors added such an analysis, I would have increased my score quite a bit. As things stand, I will only raise it to 6 (Though @meta-reviewer: if the paper hinges on my review, I'd rather have it accepted, and would be willing to increase my score to 7. I might do so anyways in a couple of days after I thought this through).

---

> > > > > ### Author Response · Authors · 2020-11-20
> > > > > **Interpreting behaviour of different modules.**
> > > > >
> > > > > Hello,
> > > > >
> > > > > We thank the reviewer for their thoughtful response and increasing their score. We ran more experiments to directly address your concerns about what different modules are doing.
> > > > >
> > > > > *Env Setup*: We take a trained agent on the MemoryS13Random Environment from the MiniGrid, which is a particularly interesting environment in which the agent gets to see an object in the beginning of the episode. The agent then has to navigate through a narrow hallway and has to match the object initially shown with the correct object at the end of the hallway which splits into two paths. An efficient strategy to navigate in this environment is to get the object shown in the beginning in the field of view and match it with the right object by navigating to the end of the hallway split.
> > > > >
> > > > > - In order to analyze the behavior of the modules, we analyze  the activity patterns of all the five modules over 20 episodes, along with the incoming visual observations, (please refer to Appendix Fig. 11 and Fig 12, and with an ablation discussion under subsection: “What are the Modules Learning” (Page 8) in Section 4.4 in the main paper).  We track the agent’s behavior towards the end of the episode when the agent  is about to reach the goal. Specifically, we found that module 5 always becomes active when the agent is about to reach the goal towards the end of the episode, and module 2 gets activated when the agent reaches the junction point (i.e.,  end of the hallway split and around the last 2-3 steps of the episode). We note that this behavior is consistent across all episodes, showing a particular trend in the module activations in certain parts of the state space.
> > > > >
> > > > > - We also study what happens if we "shut off" one of the modules which may be relevant for the processing of the visual information (refer to page 8 in section 4.4 in the main paper).  For an agent trained on a setting consisting of 5 modules (with 3 active modules), during testing we "shut" off one the modules which win the competition. We find that "shutting" off the winning module degrades performance indicating how different modules may have developed certain specializations which may be necessary for good performance. More concretely, with all three active modules, the agent completes 20 episodes in 192 frames, while keeping only two modules active takes longer (243 frames), and only one active modules takes much longer (2179 frames), indicating that each module plays a certain important role throughout the interaction span.
> > > > >
> > > > > We thank the reviewer again for this suggestion as this study helped in providing some very useful insights about the behaviour of the different modules. We appreciate if the reviewer has any  further feedback. Let us know if we can do something more to improve your score.
> > > > >
> > > > > Thank you!

---

### Official Review · AnonReviewer5 · 2020-11-08
**Review of "Meta Attention Networks: Meta-Learning Attention to Modulate Information Between Recurrent Independent Mechanisms"**

**Rating:** 7
**Confidence:** 4

**Review:**


Summary:

The authors present an approach that leverages inductive biases for knowledge decomposition and relational reasoning with the aim of generalizing over out-of-distibution tasks in RL with stationary rewards.  They propose a meta-learning training framework where parameters of the network are divided into two subsets where one set is learned with fast updates and the other with slow updates capturing the slow changing dynamics of the environment.

They also propose an architecture that includes N modular components in the form of recurrent independent mechanisms (RIMS, Goyal et al. 2019) and two multi-head dot-product attention components: input attention for selecting k modules and communication attention between the selected modules.  The attention parameters are updated slowly and model the stable dynamics of decomposing task input information as meta-learning parameters while the modules are updated more frequently while the attention dynamics are held constant.  To train the policy and value function the authors chose to use Proximal Policy Optimization (PPO).  Here the value head is updated slowly with the attention meta-parameters while the policy is updated in sync with RIMs.

This approach is evaluated against GridWorld and BabyAI suite tasks where the task complexity is parameterized and involve navigating in grid worlds with partial observability conditional on input instructions.  Generalization performance is measured on more complex out-of-distibution tasks and the authors also carry out ablation study on the meta-learning setup as well as analysis on the learned value function and module communication sparsity.

Strengths & Weaknesses:

Meta-learning and factored representations are important areas of study if we are to build RL agents that can generalize well and learn new tasks approaching the sample efficiency or small amounts of data required, for instance, by humans.  Therefore it is key to build meta-learning into RL agents and so this work I believe is well motivated and aims to contribute to a crucial area of study.  The work here seems to be largely inspired by Bengio et al. (2019) in learning via a meta-transfer objective among causal entities where the authors use meta-learned multi-head attention mechanisms facilitating the interaction dynamics of a network where the modular components are realized as RIMs.  While this work bears some strong similarity to this work I believe that there are some key differences in the operation of the two attention mechanisms, leveraging the  composability of RIMs and learning framework that make it a novel contribution.

I think there are a few things in the paper that could benefit from more detail, for instance, it would be helpful to provide the definition of recurrent dynamics D_j and how the attention, RIMs and RL all connect together quantitatively.  Also a bit more definition around the tasks in the main paper could be helpful.  I realize fitting things in can be tricky but even a small bit more description could help.

Is it hard to tune the loop sizes over the meta-sequence learning?  I think the impact of this work depends on this and it could be nice to see some analysis or get a high level sense of how different loop sizes affect learning.

Overall I think the results are good and the analyses are helpful in understanding the utility of the approach over the baselines in terms of sample efficiency and generalization performance over more complex tasks.  Another baseline that is able to meta-learn some parameters but without a modular architecture might have been a good point of comparison.  Although, the ablation study with the modified LSTM is a simple version of this it seems.  The analyses of the value function and module sparsity are good and I think help the overall argument.

The tasks evaluated on are GridWorld and the BabyAI.  These tasks are fairly simple toy tasks but are well suited to demonstrate the power of the model to a limit.   It would be helpful to see the approach evaluated against more complex environments where factorizing representations is more difficult to noise and complexity in the task distribution.  This might help better illustrate the horizons of this area of work.

Finally, the paper is clearly written and well structured.

Recommendation:

I think overall this is a nice paper and would have no problem to see it accepted.  I think the work could be more ambitious but I believe that the authors have illustrated that this is a potentially useful innovation for RL.

---

> ### Author Response · Authors · 2020-11-13
> **Clarifications and Extra baseline**
>
> We thank the reviewer for their constructive feedback. We are happy that reviewer find the paper clearly written, and well structured. We conduct more experiments in order to answer the following questions.
>
> > "it would be helpful to provide the definition of recurrent dynamics D_j and how the attention, RIMs and RL all connect together. "
>
> RIMs consist of an ensemble of independent recurrent modules, which have dynamic connections facilitated by two sets of attention mechanisms. We lay out the different components in Fig 1 in the paper, and also explain them below. The main characteristics of the recurrent independent modules are:
>
> -  The underlying dynamics for each recurrent module is a regular LSTM or GRU, and the parameters are independent for each module. Say we have a total of n modules, then the dynamics of a module j is referred to as D_j, j=1,...,n i.e., different modules have different parameters.
> -  Given an ensemble of “n” modules, only those modules get to update their internal dynamics which are most relevant to the incoming input at  timestep t.
> - This selection of relevant modules is done by an (input) attention mechanism in which a query is generated by each module, and all “n” modules compete to attend to this input. The modules which have the highest attention scores win, and get activated.
> -  These activated modules can query all other modules (active or not) to read any relevant contextual information through a second attention mechanism (i.e., communication attention). However, only these activated modules are updated, while the inactivated ones keep their original internal states.
>
> The overall connection with the RL setup is as below:
>
> - As an agent interacts with the world, i.e. the BabyAI and MiniGrid environments, it receives an input sequence of a partially-observed egocentric view of the world. These RGB images are processed through an encoder.
> - The processed observations are then passed to a recurrent network which can learn from the sequential structure of the observations. This is the part where different recurrent networks can be chosen, e.g. LSTM, GRU, RIMs and so on.
> - The output from this recurrent part is then fed to separate policy and value function heads, which learn the policy and the value function respectively.
>
> We hope that this clarifies the question.
>
> > "Also a bit more definition around the tasks in the main paper could be helpful. "
>
> Thank you for this suggestion. We agree that a brief description of the tasks can be useful. We provide more details for each of the tasks, as well as some screenshots in Appendix A.2: Environments: Minigrid And BabyAI to provide a better context.
>
> > " Is it hard to tune the loop sizes over the meta-sequence learning?"
>
> This is a particularly interesting question and in order to evaluate the impact of the sizes of the sequences in the meta learning loop, we did some experiments with different variations of the size of the meta-sequence as an ablation study. As depicted in Fig 6.a., we found that a longer meta-episode helps in faster learning. Essentially, as shown in Fig 6.a, for the curve labeled “shorter_meta_ep, n=5, k=3”, we reduced the length of the meta-sequence to 2 times the length of the sequence in the inner loop, as opposed to the current setting of 4 times, and found that the setting of a longer meta sequence (i.e. of 4 times) performed better. However, note that in both situations, the meta learning setting outperformed the baselines. We believe this is the case because as intended in the current setup, we are trying to capture long-term stable patterns in the outer loop, and providing longer sequences in the slower loop helps with this objective.
>
> "Another baseline that is able to meta-learn some parameters but without a modular architecture might have been a good point of comparison."
>
> Thank you for this suggestion. We agree that evaluating a non-modular architecture in the current setup would be beneficial. As already pointed in the question, we did a simple version of this by learning an LSTM in a meta-learning fashion, and found improvements, as shown in Fig 5b for curve labeled “meta-lstm”, emphasizing the importance of the meta-learning setup.
>
> "It would be helpful to see the approach evaluated against more complex environments where factorizing representations is more difficult to noise and complexity in the task distribution."
>
> We agree that expanding this approach to more complex environments would be an interesting direction. Moving to multi-room and more challenging environments in the MiniGrid and BabyAI suites substantially increases the difficulty of the tasks as one needs to factor into better exploration strategies (to tackle the sparse reward problem) which is not the central theme of the current submission. However, this would be a very interesting extension of the current approach, which we would like to consider as future work.

---

### Author Response · Authors · 2020-11-25
**Common Response: Summary of our changes during the rebuttal period**

We thank the reviewers for their time and feedback on our paper. It has helped us in improving our paper. We feel like  that these additional changes address most of the concerns raised by the reviewers.

Here’s the summary of our edits:

1. **Results on more environments** - Added results from three additional environments - DynamicObstacles, LavaCrossing and CrossingS9N3. We found that the proposed method outperforms the baselines in all the three environments. Ref to Fig. 3 and Fig. 4 (on Pages 6, 7) in the main paper.

2. **Effect of reducing the learning rate of attention parameters** - Added ablation study to analyze the effect of slowing down the learning rate of the attention parameters. We found that just reducing the learning rate of the attention parameters does not do as well as the proposed method; Please see Fig. 8 (on Page 9) in the main paper.

3. **Reversing the role of fast and slow parameters** - Added ablation study to depict the role of using attention parameters as slow parameters and vice versa. This shows that using attention parameters as slow parameters performs  better as compared to the scenario where the roles are reversed. Refer to Fig. 8 (on Page 9) in the main paper.

4. **What different modules are learning** - Visualization of what different modules are learning across 20 episodes with full plot of input sequence and the module activation patterns at every timestep. This  plot shows how certain modules get activated in certain regions of the  state space, indicating specialization. Refer to  Fig. 11, and Fig. 12 in Appendix (on Pages 16, 17).

5. **Analyzing the different modules by turning them off** - Specific roles played by the modules that win the competition, by incrementally turning off the modules one by one, leading to constant degradation in performance; Refer to Fig. 9 (on Page 9).

---

### Decision · Program_Chairs · 2021-01-07
**Final Decision**

**Decision:**

Accept (Poster)

**Comment:**

The paper examines an idea that knowledge and rewards are stationary and reusable across tasks. An interesting paper that combines number of related topics (meta RL, HRL, time scale in RL, and attention), improving the speed of training.

The authors have addressed the reviewer comments, strengthening the paper. The reviewers  agree, and I concur, that the paper contributes a novel model,  valuable to the ICLR community. It is well thought-out, presented, and evaluated.